# An Analysis of Variability in the Content of Phenolic Acids and Flavonoids in Camelina Seeds Depending on Weather Conditions, Functional Form, and Genotypes

**DOI:** 10.3390/molecules27113364

**Published:** 2022-05-24

**Authors:** Danuta Kurasiak-Popowska, Małgorzata Graczyk, Anna Przybylska-Balcerek, Kinga Stuper-Szablewska, Lidia Szwajkowska-Michałek

**Affiliations:** 1Department of Genetics and Plant Breeding, Faculty of Agronomy, Horticulture and Bioengineering, Poznań University of Life Sciences, ul. Dojazd 11, 60-632 Poznań, Poland; danuta.kurasiak-popowska@up.poznan.pl; 2Department of Mathematical and Statistical Methods, Faculty of Agronomy, Horticulture and Bioengineering, Poznań University of Life Sciences, ul. Wojska Polskiego 28, 60-637 Poznań, Poland; malgorzata.graczyk@up.poznan.pl; 3Department of Chemistry, Faculty of Forestry and Wood Technology, Poznań University of Life Sciences, ul. Wojska Polskiego 75, 60-625 Poznań, Poland; anna.przybylska@up.poznan.pl (A.P.-B.); kinga.stuper@up.poznan.pl (K.S.-S.)

**Keywords:** *Camelina sativa*, bioactive compounds, antioxidative properties, phenolic acids, flavonoids

## Abstract

Camelina oil obtained from the seeds of *Camelina sativa* exhibits strong antioxidative properties. This study was based on four years of field experiments conducted on 63 genotypes of spring and 11 genotypes of winter camelina. The aim of the study was to determine the variability in the content of the selected bioactive compounds, depending on the weather conditions during the cultivation, the functional form, and genotype. The cultivation form of the genotypes analysed in our study did not exhibit significant differences in the quantitative profiles of the tested phenolic acids and flavonoids. Sinapic acid was the most abundant of all phenolic acids under analysis (617–668 mg/kg), while quercetin was the main flavonoid (91–161 mg/kg). Camelina has great potential not only for the food industry but also for researchers attempting to breed an oil plant with the stable biosynthesis of bioactive compounds to ensure oxidative protection of obtained fat.

## 1. Introduction

A well-balanced diet ensures normal body function. Among various dietary components, fats are particularly important for human health. Fats of the highest possible oxidative stability should be consumed, with the lowest possible content of secondary products of oxidation of essential unsaturated fatty acids [1]. However, the processes of the oxidation of fats, especially vegetable fats, are initiated during their production. Unfavourable changes in oils may be initiated in oilseeds, but they are inevitable during the production of oils. Cold-pressed oils have high oxidation stability due to the presence of antioxidants [2]. The oil-refining process reduces the contents of antioxidants: tocopherols, phospholipids, carotenoids, sterols, and polyphenols [3]. During the desludging and deacidification of rapeseed oil, the contents of phospholipids and tocopherols are reduced by about 85% and 42%, respectively. The loss of sterols does not have a major influence on the oxidative stability, because only ∆-5-avenasterol has antioxidative properties [4]. The processes of oil bleaching at 175–225 °C and deodorisation at 240–270 °C result in the formation of certain amounts of fatty acids containing systems of two and three conjugated double bonds, which are more susceptible to oxidation [5]. Cold-pressed oils are subjected to short-term processing at low temperatures not exceeding 40 °C, which guarantees that seeds preserve their health-promoting and gustatory properties. Among the oils produced with traditional methods, camelina oil is gaining increasing importance, because it exhibits antioxidative properties [6,7]. It is obtained from the seeds of *C. sativa*. Camelina oil has a characteristic aftertaste of onion and mustard seeds. Its colours range from golden (straw) to red and brown. Due to the high content of antioxidants, it can be stored for a long time, even 6 month and longer [8]. As camelina oil has a rich composition of bioactive compounds, it may have cardioprotective properties and positively influence the lipid profile [9,10]. As there have been few studies on this subject, and due to the potential beneficial properties of camelina oil, further analyses are necessary [11]. In their earlier studies, these authors analysed the seeds of camelina in the presence of flavonoids and phenolic acids and found the occurrence of flavonoid aglycones (apigenin, catechin, kaempferol, luteolin, naringenin, quercetin, rutin, and vitexin) and phenolic acids (4-hydroxybenzoic, chlorogenic, gallic, protocatechuic, syringic, vanillic, vanillin, caffeic, ferulic, *p*-coumaric, sinapic, and t-cinnamic) [6]. Other authors have studied the contents of phenolic compounds in the camelina seeds they marked: ellagic acid, protocatechuic acid, p-hydroxybenzoic acid, sinapic acid, salicylic acid, catechin, rutin, quercetin, and quercetin glucoside [12]. The conclusions about the factors influencing the chemical compositions of seeds should be based on a series of long-term studies, in which the plants under observation should grow under the same agrotechnical conditions to minimise the variables.

Therefore, this research focused on the analysis of selected bioactive compounds with antioxidative properties. This study was based on four years of field experiments conducted on 63 genotypes of spring camelina and 11 genotypes of winter camelina (Appendix A). The aim of this study was to determine the variability in the contents of selected bioactive compounds, depending on the weather conditions during the cultivation, the functional form, and the genotype.

## 2. Results and Discussion

### 2.1. Weather Conditions

Similar to in other oilseeds, the profiles of bioactive compounds in camelina seeds are conditioned by various factors [13]. One of them is the variability of the weather conditions, which acts as a stress factor responsible for the intensified biosynthesis of the antioxidants analysed in this study. Their presence counteracts the effects of oxidative stress caused by abiotic factors, such as temperature fluctuations and variability in the amount of rainfall. The situation is complicated by the fact that the same variety grown under similar agricultural conditions may differ significantly in the content of bioactive compounds [14]. A study on *Arabidopsis* showed an increase in the aliphatic glucosinolate and flavonoid levels as a result of waterlogging and drought stress. The optimal temperature during the cultivation of broccoli activated glucosinolate biosynthesis and resulted in higher bioactivity values [15]. Higher temperatures caused the formation of greater amounts of phenolic compounds in the Parthenon broccoli cultivar [15]. The authors of this study checked whether such a variability also applied to *C*. *sativa* of the Brassicaceae family, which is native to Eastern Europe and Western Asia [16]. This annual oil plant easily adapts to various climatic and soil conditions, and it is highly resistant to diseases and pests [17,18,19]. Camelina is the least sensitive of all cruciferous plants to temporary soil water deficits. So far, studies on the stability of *C. sativa* characteristics have not provided unambiguous results, and the vast majority of them have been conducted on spring cultivars of this species [20]. Chemical analyses of *C. sativa* usually concentrate on its fatty acid profile. Our study, spanning a period of four years, was conducted on 63 spring and 11 winter genotypes of camelina. It showed that the weather conditions did not affect the quantitative profile of fatty acids [11]. On the other hand, other authors have observed that, during the filling of camelina seeds, the content of α-linolenic acid in *C. sativa* regions with lower temperatures (Canada and Poland) was higher than in the warmer Mediterranean climate [21].

During the current research (2016–2019), the weather conditions varied considerably (Figure 1). The total rainfall between April and July amounted to 312 mm in 2016, 266 mm in 2017, 268 mm in 2018, and only 132 mm in 2019, whereas the long-term average (1956–2009) for this period was 235 mm. The average long-term air temperature was 8.2 °C for April, 13.5 °C for May, and 16.8 °C for June. During the current study, there were much higher air temperatures in April (14.4 °C) and May (18.4 °C) 2018. The air temperature in June was 21 °C in 2016, 19 °C in 2017, 19.7 °C in 2018, and 23.7 °C in 2019. In all years of the current research, the temperature in June was much higher than the long-term average temperature. The average air temperature in July ranged from 19.5 °C to 21 °C. The highest air temperatures in spring were recorded in 2018. During the growing season of 2019, there was very low rainfall and the highest temperatures. During the four years of the field research, there were various weather conditions, which may have affected the contents of bioactive compounds in camelina seeds. Therefore, changes in the contents of polyphenolic acids and flavonoids during the four-year research period were analysed.

The plots for the polyphenols and flavonoids are shown in Figure 2a,b. The indicators of one variable (acids) are on the OX-axis. A separate curve was made for the means of each level of the second variable, i.e., variety. Each curve consists of values of acid contents determined for the studied years. Significant interactions between variables and years impact on the changes in this picture, and the curves should not parallel, as they intersected at different angles. In the presented picture (Figure 2), we deal with the opposite case. The polygons showing the interactions of the polyphenols and flavonoids are almost parallel. This indicates that the effects of the interactions between the types of the tested polyphenols and flavonoids and the years on the contents of the components in the plants were not significant. However, there were differences in the acid contents in various forms.

### 2.2. The Content of Polyphenols

Generally, there were no differences between the contents of individual polyphenols and the research years 2016–2019 (Appendix A). Sinapic acid was the most abundant of all the polyphenolic acids under analysis (Appendix A). The average contents of sinapic acid in the research years ranged from 1593.48 mg/kg (2017) to 1823.78 mg/kg (2018). The contents of this acid were particularly strongly diversified—diversification from the main value ranged from 617 mg/kg to 668 mg/kg. Similarly, other authors noted that the major phenolic acid in camelina was sinapic acid (2532.78 mg/kg sample) [22]. A small amount of gallic acid was found in the insoluble-bound fractions of both samples, i.e., 8.86 and 4.79 mg/kg sample, respectively.

During the study, there were also high levels of caffeic, *t*-cinnamic, and 4-hydroxybenzoic acids. The average content of caffeic acid ranged from 149 mg/kg (2017) to 202 mg/kg (2016), *t*-cinnamic acid from 147 mg/kg (2018) to 166 mg/kg (2016 and 2019), and 4-hydroxybenzoic acid from 149 mg/kg (2016) to 180 mg/kg (2017). The contents of the syringic, chlorogenic, and ferulic acids varied from 58 mg/kg (chlorogenic acid in 2018) to 105 mg/kg (syringic acid in 2018). The contents of the other polyphenols (gallic, *p*-coumaric, protocatechuic, vanillic acids, and vanillin) did not exceed 40 mg/kg.

The analysis of the contents of the flavonoids showed no differences between individual years (2016–2019) during the period under study (Appendix A). Among the flavonoids under analysis, quercetin was characterised by the highest and most variable content; on average, it ranged from 91 mg/kg in 2019 to 161 mg/kg in 2017. The camelina seeds also had a relatively high content of apigenin; it ranged from 79 mg/kg to 102 mg/kg. The seeds had a similar content of luteolin and naringenin—from 58 mg/kg to 84 mg/kg—and a similar content of catechin, kaempferol, and vitexin—from 21 mg/kg to 35 mg/kg.

There have been earlier studies on the contents of the abovementioned bioactive compounds in camelina seeds. The results depended strictly on the cultivar, the extraction method, and growing conditions. For example, some studies used HPLC-DAD MS/MS to analyse the genotypes of *C. sativa* and found that they contained 1954.55 mg/kg of phenolic acids [22]. The total phenolic contents (free, esterified, and insoluble, bound phenolic extracts) registered by some authors for camelina were, on average, 11.69 mg GAE/g defatted meal (4.07 mg GAE/g sample of free phenolics and 0.82 mg GAE/g sample of insoluble-bound phenolics) [22]. The total flavonoids content (TFC) in camelina was 6.81 mg CE/g defatted meal [22]. According to other authors, qualitative differences in the phenolic profiles of *Brassica* crops should be compared rather than differences in concentrations, because different environmental conditions and analytical methods cause differences in the results [13].

A principal component analysis was applied to indicate the polyphenols with the biggest variations in the contents of acids and to indicate relations between the polyphenols and cultivars (Figure 3a,b and Figure 4a,b). The subset of principal components was chosen according to the scree plots. The analysis showed that the first principal component explained 27.3% of the total variability in the contents of polyphenols in the cultivars and was strongly correlated with four of the original variables: sinapic, caffeic, vanillin, and 4-hydroxylic acids. The first principal component increased along with the content of sinapic, caffeic, vanillin, and 4-hydroxylic acids. If one of them increased, so did the others. PC1 was the most strongly correlated with the content of sinapic acid. The second principal component explained 23.2% of the total variability in the contents of the polyphenols in the cultivars and was strongly correlated with two of the original variables: chlorogenic and vanillic acids. The third principal component explained 15.5% of the total variability of the polyphenols in the cultivars and was correlated with gallic and protocatechuic acids.

The PCA revealed correlations between the contents of the following polyphenols: caffeic and 4-hydroxylic acids, *p*-coumaric and protocatechuic acids, t-cinnamic and protocatechuic acids, sinapic, syringic acids, and vanillin, as well as chlorogenic, gallic, and ferulic acids.

The following polyphenols were the most strongly correlated with the winter cultivars: caffeic, 4-hydroxylic, *p*-coumaric, protocatechuic, t-cinnamic, sinapic, syringic, and vanillic acids. Chlorogenic, gallic, ferulic, and vanillic acids were the most strongly correlated with the spring varieties. The spring varieties formed a distinct cluster on the left, whereas the winter varieties formed a cluster on the right. The spring varieties were characterised by high contents of chlorogenic, gallic, and ferulic acids. The winter varieties had high contents of sinapic and hydroxybenzoic acids. There was a distinct division line between the spring and winter varieties along the principal component that was closely correlated with chlorogenic, sinapic, gallic, and hydroxybenzoic acids. The winter varieties differed significantly from the spring ones; they had opposite levels of significant traits.

Genotypes CSS_CAM8 and Index Seminum 144 (no. 41 and 42) had increased contents of t-cinnamic, protocatechuic, and *p*-coumaric acids, whereas the content of vanillin in genotypes 14 2 2 was higher than in the other spring varieties. Polish winter camelina mutation line 15/2/3 had greater than average contents of apigenin, kaempferol, quercetin, rutin, vitexin, hydroxybenzoic, caffeic, chlorogenic, ferulic, sinapic, t-cinnamic acids, and vanillin (almost all acids). There were similar results for winter camelina line C5 and variety Lenka.

Some genotypes shown in Figure 4a,b (PCA) are located in the area between the winter and spring groups (CPS-CAM10, CSS-CAM31, CSS_CAM8, and Index Seminum 144), as they had moderate contents of the acids under analysis.

The PCA showed that the first principal component explained 28.2% of the total variability in the contents of flavonoids in the varieties and was strongly correlated with four of the original variables, i.e., apigenin, kaempferol, catechin, and luteolin (Figure 4). The first principal component increased, along with the contents of the aforementioned sterols. If one of them increased, so did the others. PC1 was the most strongly correlated with the apigenin content. The second principal component explained 22.7% of the total variability in the contents of flavonoids in the varieties and was strongly correlated with quercetin. The third principal component explained 14.2% of the total variability of the flavonoids in the varieties and was correlated with vitexin. The contents of the flavonoids naringenin and kaempferol were also correlated.

The following flavonoids were the most strongly correlated with the winter varieties: apigenin, kaempferol, luteolin, and rutin. Quercetin and vitexin were the most strongly correlated with the spring varieties.

The spring genotypes formed a distinct cluster on the left, whereas the winter genotypes formed a cluster on the right. However, there were a few varieties that did not fit into either of these clusters. The spring varieties were characterised by a high content of vitexin, quercetin, and acids. The winter varieties had high contents of kaempferol, apigenin, luteolin, and acids.

The results showed that the spring varieties had relatively high contents of vitexin. The winter varieties differed significantly from the spring ones—they had opposite levels of significant traits.

Unlike the other winter varieties, line 14 2 2 had a lower content of the acids, which determined the winter varieties cluster. Mutation line 14/2/3 had a moderate content of the acids, which determined the winter varieties cluster.

Among the spring lines, GE2011-05, BRSCHW 30021, CSS_CAM 27, and Omskij Mestnyj had the highest contents of quercetin, catechin, and acids. Unlike the other spring lines, Giessen#3, CSS_CAM 38 and No 403 had increased contents of luteolin and rutin.

Some varieties shown in Figure 4 are located in the area between the winter and spring groups (UP 2017/02, Hoga, CSS-CAM 30 and Zavolzskij), as they had moderate contents of the acids under analysis.

According to some authors [13], the biosynthesis and concentration of phenolic compounds in Brassica species depends on genetic factors and differs between and within species. However, it primarily depends on the genotype.

Apart from the PCA, a heat map and discriminant analysis were used to indicate and explain the relationships between varieties and the contents of the bioactive compounds under analysis. The heat map shows information on the integration sites of the datasets in columns (genotypes) and different polyphenols and flavonoids in rows (Figure 5a,b).

There was no variation between the genotypes in the contents of the polyphenols. Sinapic acid had the greatest discriminatory power for the varieties (Table 1). This result was confirmed by a discriminant analysis (Figure 5a,b). The similarity dendrogram divided the genotypes into five groups, depending on their contents of polyphenolic compounds. The genotypes from groups 1 and 2 and those from groups 3–5 were at the greatest distance from each other. In groups 1 and 2, there were only winter camelina genotypes, whereas groups 3, 4, and 5 had only spring camelina genotypes. Separate cluster groups for the spring and winter forms of camelina showed that the functional form had a big influence on the contents of the polyphenols in the seeds.

Quercetin, apigenin, luteolin, and naringenin had the greatest discriminatory power for the varieties. There were no variations between the cultivars in the content of rutin. The heat map results were confirmed by a discriminant analysis (Figure 5a,b). Apigenin and kaempferol had the greatest discriminatory power (Table 1). There were five cluster groups in the similarity dendrogram. The genotypes in groups 1–3 exhibited the greatest differences from the genotypes in groups 4 and 5. Our knowledge about the collected camelina genotypes was insufficient to explain why individual genotypes belonged to specific classes. Neither the functional form nor the origin played a significant role in the dendrogram.

## 3. Materials and Methods

### 3.1. Test Material

The research was conducted on 63 genotypes of spring camelina: 44 biotypes acquired from the US National Plant Germplasm System (NPGS), Washington, D.C., USA and 19 genotypes from the collection of the Department of Plant Genetics and Breeding, Poznań University of Life Sciences, Poland (Appendix A). Eleven genotypes of winter camelina were also analysed: 4 warieties: ‘Przybrodzka’, ‘Luna’, ‘Maczuga’, and ‘Lenka’ and 7 genetically stable mutation lines obtained after the irradiation of ‘Przybrodzka’ in 1993.

### 3.2. Field Experiments

Between 2016 and 2019, during the growing seasons, field experiments were conducted at the Agricultural Research Station in Dłoń, Poznań University of Life Sciences, Poland (51°41′37″ N 17°04′06″ E). The experiment was conducted in a randomised block design with three replicates.

Data on the average monthly temperatures and rainfall, measured according to the WMO guidelines between 2016 and 2019, were provided by a Vantage Vue 6357 UE 9 meteorological station (Davis Instruments, Hayward, USA) located about 400 m from the experimental field.

### 3.3. Analysis of Phenolic Compounds

Phenolic compounds in samples were analysed after alkaline and acidic hydrolysis [23,24]. The analysis was conducted with an Acquity UPLC H-class system equipped with a Waters Acquity PDA detector (Waters, Milford, Ma, USA). Chromatographic separation was conducted on an Acquity UPLC^®^ BEH C18 column (100 mm × 2.1 mm, particle size 1.7 μm) (Waters, Dublin, Ireland). Gradient elution was conducted with the following mobile phase composition: (A) acetonitrile with 0.1% formic acid and (B) 1% aqueous formic acid mixture (pH = 2). The concentrations of phenolic compounds were determined with an internal standard at wavelengths λ = 320 nm and 280 nm. Compounds were identified by comparing the retention time of the peak under analysis with the retention time of the standard and by adding a specific amount of the standard to the samples being analysed and repeating the analysis. The detection level was 1 μg/g. The retention times of the assayed compounds were as follows: kaempferol—6.11 min, gallic acid—8.85 min, vanillic acid—9.71 min, luteolin—11.89 min, protocatechuic acid—12.23 min, vanillin—14.19 min, apigenin—16.43 min, catechin—18.09 min, 4-hydroxybenzoic acid—19.46 min, chlorogenic acid—21.56 min, caffeic acid—26.19 min, syringic acid—28.05 min, naringenin—31.22 min, vitexin—35.41 min, rutin—38.11 min, quercetin—39.58 min, *p*-coumaric acid—40.20 min, ferulic acid—46.20 min, sinapic acid—48.00 min, and *t*-cinnamic acid—52.40 min. The recovery rates of the phenolic compounds were as follows: kaempferol—86 ± 5.3%, gallic acid—92 ± 4.4%, vanillic acid—79 ± 8.5%, luteolin—96 ± 2.7%, protocatechuic acid—90 ± 4.8%, vanillin—88 ± 5.1%, apigenin—93 ± 3.8%, catechin—89 ± 5.7%, 4-hydroxybenzoic acid—96 ± 3.78%, chlorogenic acid—92 ± 2.8%, caffeic acid—86 ± 6.7%, syringic acid—94 ± 3.9%, naringenin—88 ± 4.8%, vitexin—95 ± 3.8%, rutin—93 ± 4,9%, quercetin—97 ± 1.9%, *p*-coumaric acid—89 ± 3.6%, ferulic acid—91 ± 4.9%, sinapic acid—94 ± 5.1%, and *t*-cinnamic acid—97 ± 2.9% [24].

### 3.4. Statistical Analysis

The samples were grouped in box plots. They showed variations in the samples of the considered statistical population: the contents of the acids without making any assumptions about the underlying statistical distribution. In order to show the relationship between the considered varieties and the contents of the acids, an interaction plot was presented. This plot displayed levels of one factor on the x-axis (acids) and a separate line for each level of another factor (varieties). In order to visualise the contents of the polyphenols and flavonoids in the varieties, heat maps combined with clustering methods grouping varieties and acids together were given. Each row in the heat maps represented an acid, and each column represented a variety. The colour and intensity of the boxes showed changes in the acid contents. In order to summarise and to visualise the information in a dataset containing individuals described by multiple interrelated quantitative variables, a principal component analysis was conducted. The PCA enabled the extraction of important information from a multivariate data table. This information was expressed as a set of few new variables. These new variables corresponded to a linear combination of the originals. The aim of the PCA was to identify directions (or principal components) along which the variation in the data were the maximal. A linear discriminant analysis was conducted as the next step. The LDA is closely related to the PCA and attempts to model the differences between the classes of data.

## 4. Conclusions

Our long-term study showed that the weather conditions significantly influenced the contents of the bioactive compounds under analysis. It is a natural response of plants to changeable weather conditions causing oxidative stress. The presence of antioxidants in camelina seeds ensures the oxidative stability of the fat they contain, which is important for the quality of camelina oil. On the other hand, when plants counteract the effects of oxidative stress, they maintain homeostasis and good condition; thanks to which, they yield effectively and provide high-quality products.

The statistical analyses conducted in our study enabled the identification of the most important antioxidants from the group of phenolic acids and flavonoids contained in camelina seeds. The cultivation form of the genotypes analysed in our study did not exhibit significant differences in the quantitative profiles of the tested phenolic acids and flavonoids. However, this dependency could be seen at the variety level. The compounds with the greatest discriminatory power were determined for the varieties under study. This means that individual camelina genotypes have a thorough biochemically significant influence on the resistance processes, which they control by means of the intensified biosynthesis of selected bioactive compounds. This information is particularly valuable, because so far, there has been no study on such a large genetic population of oil plants grown under the same agrotechnical conditions. The growing conditions excluded most of the factors influencing the profiles of the bioactive compounds, and the only variable factors were the weather conditions, functional form, and genotype. Our study showed that camelina has great potential not only for the food industry but also for researchers attempting to breed an oil plant with the stable biosynthesis of bioactive compounds to ensure oxidative protection of the obtained fat.

## Figures and Tables

**Figure 1 molecules-27-03364-f001:**
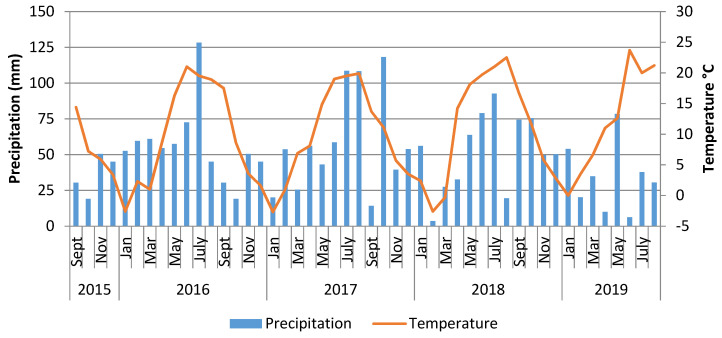
Distribution of rainfall and mean temperature during the field experiment in Dłoń, Poland.

**Figure 2 molecules-27-03364-f002:**
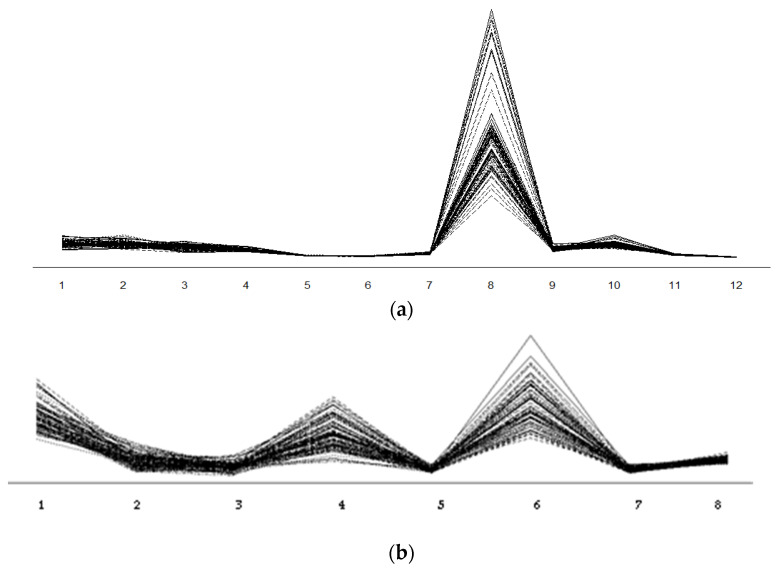
The plot between polyphenols (**a**) 4-hydroxybenzoic acid (1), Caffeic acid (2), Chlorogenic acid (3), Ferulic acid (4), Gallic acid (5), *p*-Cumaric acid (6), Protocatechuic acid (7), Sinapic acid (8), Syringic acid (9), *t*-Cinnamic acid (10), Vanillic acid (11), Vanillin acid (12) and flavonoids (**b**) Apigenin (1), Catechin (2), Kampferol (3), Luteolin (4), Naringenin (5), Quercetin (6), Rutin (7), Vitexin (8) and *C. sativa* genotypes.

**Figure 3 molecules-27-03364-f003:**
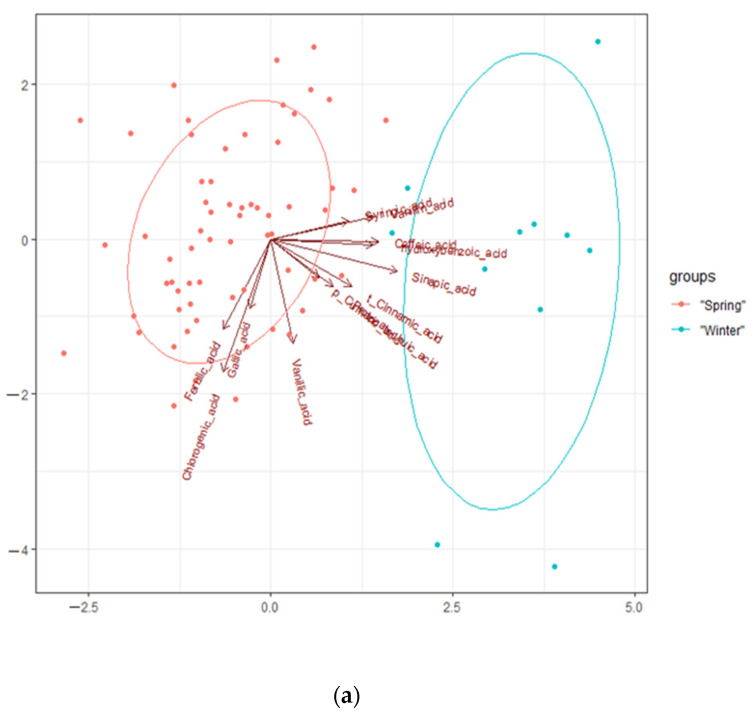
PCA for the phenols and genotypes. (**a**) PC1 (27.3% explained variance) and PC2 (23.2% explained variance) and (**b**) PC1 (27.3% explained variance) and PC3 (15.5% explained variance).

**Figure 4 molecules-27-03364-f004:**
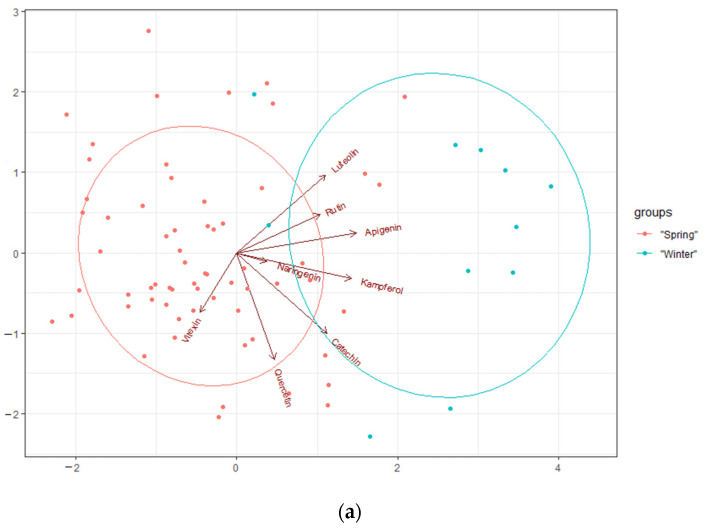
PCA for the flavonoids and genotypes. (**a**) PC1 (28.2% explained variance) and PC2 (22.7% explained variance) and (**b**) PC1 (28.2% explained variance) and PC3 (14.2% explained variance).

**Figure 5 molecules-27-03364-f005:**
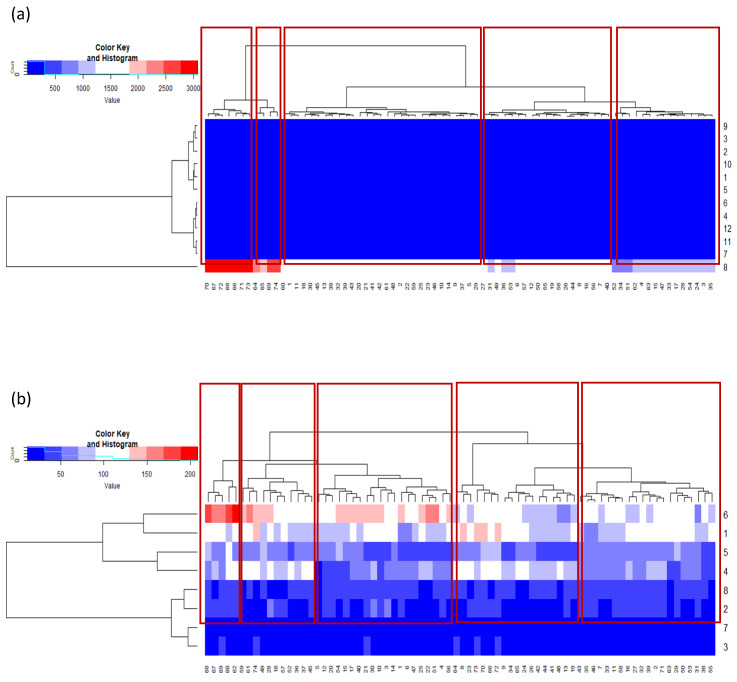
The heatmap and dendrograms for spring and winter genotypes of *C. sativa* and phenols contents: (**a**) (1) Caffeic acid, (2) Chlorogenic acid, (3) Ferulic acid, (4) Gallic acid, (5) 4-hydroxybenzoic acid, (6) *p*-Cumaric acid, (7) Protocatechuic acid, (8) Sinapic acid, (9) Syringic acid, (10) *t*-Cinnamic acid, (11) Vanillic acid, (12) Vanillin acid, and flavonoids contents and (**b**) (1) Apigenin, (2) Catechin, (3) Kampferol, (4) Luteolin, (5) Naringenin, (6) Quercetin, (7) Rutin, and (8) Vitexin.

**Table 1 molecules-27-03364-t001:** Discriminant analysis for phenols and flavonoids.

	Lambda-Wilks	R^2^
**phenols**		
Sinapic acid	0.370077	0.136062
Protocatechuic acid	0.144588	0.201786
Syringic acid	0.127974	0.070381
Caffeic acid	0.130240	0.070799
9-hydroxybenzoic acid	0.128604	0.101760
Chlorogenic acid	0.126685	0.311918
Ferulic acid	0.123810	0.159464
**flavonoids**		
Apigenin	0.539560	0.031118
Kaempferol	0.572799	0.029863
Vitexin	0.467732	0.026047
Rutin	0.438533	0.0211658

## Data Availability

Not applicable.

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
