# Peer review of "An Analysis of Variability in the Content of Phenolic Acids and Flavonoids in Camelina Seeds Depending on Weather Conditions, Functional Form, and Genotypes"

_molecules, 2022, doi:10.3390/molecules27113364_

Round 1

Reviewer 1 Report

The subject is of interest, and the experiments quite well conducted, however there are many typing errors and English mistakes to correct before this paper could be accepted for publication in Molecules

Line 23-24: setence unclear, please rephrase it for more clarity ("flavonoids quercetin")

"fenols" should be phenols

"Protocate"= protocatechuic acid?

Table 1 "acid" is missing (eg caffeic acid, not just caffeic)

katechin should be catechin

kempfero should be kaempferol

Figure 3 is difficult to read. Its quality is low. Idem for Figure 4.

p-Cummaric should be p-coumaric acid (with p in italics)

4hydroxylowy ?

pCinnamic should it be t-cinnamic acid (with t in italics)?

Reviewer 2 Report

At the beginning I have one important questions for authors to be clarified:

  1. In supplementary file (Table S1) I have found data that some of seed samples were labeled as "Former Soviet Union" so what that does mean? Were these seed samples collected during the Soviet era? Because then that means they are over 30 years old? In that case, how we can be sure that their chemical composition is not influenced by their significant age?  If not, and if they were collected in the previous 4 years when the study was conducted, how can they be marked like this? If I understood correctly from the Lines 295-296 you actually sowed these seeds and did not do their analysis? In addition, what means this "obtained after irradiaton of..." Please clarify this issues.
  2. Since you are talking here about seed as good source of oil and since in Introduction section a significant part is related to the lipophilic compounds like sterols, fatty acids, etc. why authors did not analysed them in samples? In addition, your Introduction section does not contain any data about phenolics in your seed and the whole Manuscript is committed to their analysis. So, this is contradictory and must be corrected/explained in revised form.

All other comments are made with an adequate Line number(s) from text in order to facilitate tracking.

Line 17: Delete "it". It is surplus here.

Line 24: Incomplete sentence, something is missing here like for instance: "... while quercetin was the main flavonoid (91 - 161 mg/kg)". Please check/correct/ clarified.

Lines 42-43: Provide reference(s) for given statement.

Line 57: I do not understand what is "functional form" here? Please clarify and define it.

Line 67: I do not understand why/how this Table 1 is part of Introduction section? This is statistical data so it should be given after some chemical composition data. In addition, there is a typo in the Table 1. Term "fenol" is incorrect. It must be replaced with "phenolics". Also, all names in the table are incorrect. Missing word "acid" after all phenolic acids. Also, what is "Protocate"? It should be, I suppose, protocatechuic acid? In addition, flavonoid should be "kaempferol" Correct.

Line 84: Suggest to authors to introduce here abbreviated Latin name i.e "C. sativa" since you already give the full name in Introduction section. Please apply this through a whole Manuscript.

In addition, according to the newest botanical rules Latin names for families should not be written in Italic style anymore. So, it should be Brassicaceae here. Please correct.

Line 93: typo  - split start of the next sentence from reference 10 here with a space. In addition, reference Zanetti et al. should be given as numerical citation or if it is actually reference no. 19 mentioned in the Line 95 than this year "(2017)" is completely surplus here. Check/correct.

Line 96: In scientific language there is no "I", "We", "our", etc. Please avoid to use this phrases as much as it is possible. Suggest to replace here "our" with "current".

Line 100: The same as previous.

Line 102: The same as previous.

Lines 129-135: I completely do not understand this Figure 2 and your interpretation of it. You are talking about some "interactions between phenolics and years" and where are the years on Figure 2? From Figure 2 I only can conclude what was the main phenolic acid / flavonoid in your samples. In addition, here you are also talking about some "sterols" (Line 134) but we can not find any results for these compounds anywhere in the text? Also, there are no any description for methods for sterols determination in Material and methods section. This issue must be correct or clarified in revised file.

Line 142: The same comment as in the Line 93 for given reference in the text. Check/correct.

Lines 140, 142, 143 and 145: Please check the given units? Is it ok that some are mg/kg and some are µg/g?

Line 151: In Organic chemistry letter "p" as prefix for "para" should be given in Italic style. Correct. The same issue should be corrected in the Lines 202, 206 and 218.

Line 162: The same comment for reference as in the Line 142. Also, correct term to be "HPLC-DAD" not HPLC_DAD".

Line 165: Suggest to replace term "received" with "registered" here. Also, the same comment for reference a sin the Line 162.

Line 167: It should be just "flavonoid" here not in plural. Correct.

Line 168: Again check reference given in the text.

Lines 181 and 183: Again, here some "sterols" are mentioned and we have discussion about phenolics here not about sterols? Please check/correct/ clarified.

Line 191: "phenolics" not "fenols" here. Correct.

Lines 213-214: This is the third repetition of the same sentence that was given earlier. Check/delete/correct.

Line 257: Again, issue with reference given in the text. Check/correct.

Lines 306-307: References should be numerically cited in the text here not by the names of authors. Correct.

Author Response

"Please see the attachment

Round 2

Reviewer 1 Report

The correction have been made by the authors.

Reviewer 2 Report

I have no further comments.